# Impacts of Employment Status, Partnership, Cancer Type, and Surgical Treatment on Health-Related Quality of Life in Irradiated Head and Neck Cancer Survivors

**DOI:** 10.3390/cancers16193366

**Published:** 2024-10-01

**Authors:** Ching-Rong Lin, Tsung-Min Hung, Eric Yi-Liang Shen, Ann-Joy Cheng, Po-Hung Chang, Shiang-Fu Huang, Chung-Jan Kang, Tuan-Jen Fang, Li-Ang Lee, Chih-Hung Chang, Joseph Tung-Chieh Chang

**Affiliations:** 1School of Nursing, College of Medicine, Chang Gung University, Taoyuan 333, Taiwan; ma0402@gap.cgu.edu.tw; 2Department of Radiation Oncology and Proton Therapy Center, Chang Gung Memorial Hospital at Linkou and Chang Gung University, Taoyuan 333, Taiwan; min7363@cgmh.org.tw (T.-M.H.); pts@cgmh.org.tw (E.Y.-L.S.); annjoycheng@gap.cgu.edu.tw (A.-J.C.); 3Clinical Metabolomics Core Lab, Chang Gung Memorial Hospital at Linkou, Taoyuan 333, Taiwan; 4Department of Medical Biotechnology and Laboratory Science, College of Medicine, Chang Gung University, Taoyuan 333, Taiwan; 5Graduate Institute of Biomedical Sciences, College of Medicine, Chang Gung University, Taoyuan 333, Taiwan; 6Department of Otorhinolaryngology, Head and Neck Surgery, Chang Gung Memorial Hospital and Chang Gung University, Taoyuan 333, Taiwan; bc1766@cgmh.org.tw (P.-H.C.); bigmac@cgmh.org.tw (S.-F.H.); keny@cgmh.org.tw (C.-J.K.); fang3109@cgmh.org.tw (T.-J.F.); 5738@cgmh.org.tw (L.-A.L.); 7Graduate Institute of Clinical Medical Sciences, College of Medicine, Chang Gung University, Taoyuan 333, Taiwan; 8School of Medicine, College of Life Sciences and Medicine, National Tsing Hua University, Hsinchu 300, Taiwan; 9Program in Occupational Therapy, Department of Medicine, Department of Orthopedic Surgery, Washington University School of Medicine, St. Louis, MO 63130, USA; chih-hung.chang@wustl.edu

**Keywords:** head and neck cancer, health-related quality of life, employment, partnership, cancer type, surgical treatment

## Abstract

**Simple Summary:**

This research investigates how personal and health-related factors impact the quality of life of head and neck cancer survivors treated with radiotherapy and surgery. By studying 150 patients—60 with nasopharyngeal cancer and 90 with oral cavity cancer—researchers utilized a specific survey to assess the effects of various elements, including cancer type, age, gender, relationship status, education, and employment. Findings revealed that cancer type, treatment history, and employment status significantly influenced quality of life. Patients with nasopharyngeal cancer reported better social and functional outcomes than those with oral cavity cancer. Additionally, unemployment was correlated with lower quality of life, while having a partner and undergoing certain surgeries positively impacted outcomes. These insights can aid research efforts to better support cancer survivors and customize treatment approaches for improved well-being.

**Abstract:**

**Objectives**: This study aimed to examine the relationship between health-related quality of life (HRQoL) and sociodemographic and clinical variables in survivors of head and neck cancer (HNC) treated with radiotherapy, with or without surgery. **Materials and Methods**: HRQoL was measured using the functional assessment of cancer therapy—head and neck (FACT-H&N) in a cross-sectional survey involving 150 patients. Of these, 60 had nasopharyngeal cancer (NPC), treated exclusively with radiotherapy, while 90 had oral cavity squamous cell cancer (OSCC), undergoing radical surgery followed by adjuvant radiotherapy. Key variables included cancer type, age, gender, partnership status, education, and employment, with additional clinical variables assessed in patients with OSCC. Statistical analyses included multiple regression, ANOVA, and t-tests to explore relationships between variables and HRQoL. **Results**: Cancer type, surgical treatment, and employment status emerged as significant independent predictors of HRQoL in HNC patients. Patients with NPC reported better HRQoL on three FACT-H&N subscales—social/family well-being, functional well-being, and additional concerns—compared to patients with OSCC. Unemployed individuals exhibited lower HRQoL on four subscales. In patients with OSCC, partnership status and segmental mandibulectomy were found to predict HRQoL independently. **Conclusions**: This study concludes that cancer type, surgical intervention, and employment status notably influence HRQoL among HNC patients undergoing radiotherapy. In addition, partnership status is a key factor affecting HRQoL in patients with OSCC.

## 1. Introduction

Head and neck cancer (HNC) presents a prevalent oncological challenge in Taiwan, notably characterized by nasopharyngeal carcinoma (NPC) and oral cavity squamous cell cancer (OSCC), which are the predominant subtypes. The distinctive anatomical localization and pathophysiological characteristics of these malignancies necessitate different therapeutic approaches. NPC is primarily managed with radiotherapy, either with or without chemotherapy, while OSCC often requires aggressive surgical intervention. Such therapeutic differences invariably translate into varying impacts on health-related quality of life (HRQoL) [1,2,3]. Notably, these malignancies disproportionately afflict middle-aged men, a demographic typically at the peak of their productive years, rendering the socioeconomic consequences profound for both patients and their families, consequently influencing overall HRQoL [4,5,6,7,8,9].

In the psychosocial domain, various factors, including economic status, employment status, and social support networks, profoundly influence HRQoL. Family caregivers play an important role in providing companionship, contributing significantly to the patient’s well-being. As cancer transitions into a chronic condition with advancements in medical technology, the focus on HRQoL becomes very important [10]. While an extensive body of literature examines the influence of cancer type and treatment modality on HRQoL, the impact of individual psychosocial status remains underexplored. Factors such as age, gender, economic status, and relationship status intricately intertwine with HRQoL, with younger age and female gender often being associated with heightened emotional distress post-cancer diagnosis. Moreover, the interplay between physical health, work status, and partnership dynamics as determinants of HRQoL warrants further investigation [11,12,13].

Clinical factors also significantly influence HRQoL, as treatment modalities like radiotherapy, chemotherapy, and surgery have distinct side effect profiles that impact physical function and overall well-being. Notably, head and neck surgeries, in particular, encompass diverse approaches with varying degrees of functional impairment, thereby exerting differential effects on HRQoL. Understanding these multifaceted influences is crucial for developing comprehensive strategies to optimize the post-treatment experience and enhance HRQoL outcomes for HNC patients.

This study aims to investigate the multifaceted influences on health-related quality of life (HRQoL) among head and neck cancer patients in Taiwan, focusing on the distinct impacts of nasopharyngeal carcinoma (NPC) and oral cavity squamous cell cancer (OSCC). Specifically, this study will examine how therapeutic differences, clinical factors, and individual demographic characteristics, including age, gender, partnership status, employment, and impact HRQoL outcomes. By exploring these determinants, this study seeks to develop comprehensive strategies to optimize the post-treatment experience and enhance HRQoL for HNC patients.

## 2. Materials and Methods

### 2.1. Design and Participants

This cross-sectional survey study was approved by the Institutional Review Board (IRB No: 102-0527B) of the medical center at Taoyuan City in northern Taiwan. Study participants were recruited from the follow-up outpatient clinics. When the patient met the inclusion criteria, they would be referred by a radiation oncologist. The inclusion criteria for participants were as follows: HNC patients who had completed the entire course of treatment, were over 20 years old, could communicate verbally, had no active disease at the time of the survey, had no major psychological disease (such as schizophrenia), and had no significant disfigurement prior to the diagnosis of HNC. After the research assistant explained the purpose of the study, participants had to agree to participate.

A total of 166 HNC patients were invited to join this study. A total of 16 patients (9.6%) declined, leaving 150 patients who agreed to participate and were enrolled. All participants were recruited from the follow-up outpatient clinics of the Department of Radiation Oncology at a medical center hospital. To ensure the study’s sample size was sufficient, a power analysis was performed using G * Power 3.17. The analysis indicated that the study had a power of 85% with an effect size set at 0.5 and a significance level of α = 0.05.

### 2.2. Variables and Measures

Before participating in the survey, each participant signed their informed consent. The sociodemographic data (age, gender, partnership, education, and employment status) were collected. Participants also completed the functional assessment of cancer therapy—head and neck (FACT-H&N) questionnaire. To ensure clarity and accuracy, at least one investigator or assistant was available to answer questions during the completion of these forms.

In the analysis of patients with OSCC, specific surgical procedures were included: whether facial skin was sacrificed, whether the mouth angle was preserved, whether a glossectomy was performed, whether an inferior maxillectomy was performed, and whether a mandibulectomy was performed. Glossectomies were further categorized into partial glossectomy (up to and including hemiglossectomy) and total glossectomy (total or nearly total). Mandibulectomies were classified as either marginal or segmental mandibulectomy.

The FACT-H&N consists of functional assessment of cancer therapy—general (FACT-G)^2^ and additional H&N cancer-specific concerns. The FACT-G is a 27-item questionnaire that includes four subscales: physical well-being (PWB, 7 items), social/family well-being (SWB, 7 items), emotional well-being (EWB, 6 items), and functional well-being (FWB, 7 items). The additional H&N cancer-specific concerns include nine items, excluding smoking and drinking items. The trial outcome index (TOI) is derived from the sum of PWB, FWB, and additional concerns subscales. The traditional Chinese version of FACT-H&N used in this study was the translated version by Chang and colleagues, which demonstrated acceptable internal consistency (Cronbach’s alpha = 0.78) [14]. Higher scores on the FACT-H&N, with reverse scoring for negatively worded items, indicate better HRQoL.

### 2.3. Statistical Analyses

Descriptive analysis was conducted on the sociodemographic and clinical variables to summarize the characteristics of the study population. Relationships between these variables and the FACT-H&N scores were evaluated using *t*-test, analysis of variance (ANOVA), and multiple regression. For the univariable analysis, t-test, and ANOVA were utilized. In multivariable analysis, a multiple linear regression model with backward selection was used to determine the independent effects of demographic and clinical variables on HRQoL. Variables were retained in the model based on their statistical significance and contribution to the overall model fit. A *p*-value of <0.05 was considered to be statistically significant. All statistical analyses were performed using SPSS version 21.0 (IBM-SPSS Inc., Chicago, IL, USA), ensuring rigorous and reliable evaluation of the data.

## 3. Results

Among the 150 enrolled HNC patients, the mean age was 50.9 (range 29–72) years old. The majority of patients were male (128, 85.3%) and had a partner (118, 78.7%). In terms of employment status, 38 patients (25.3%) were unemployed, 55 patients (36.7%) had part-time jobs, and 57 patients (38.0%) were employed full-time. Regarding the cancer type and treatment, 60 patients were diagnosed with NPC and treated with definite radiotherapy, with or without chemotherapy, but no surgery. The remaining 90 patients were diagnosed with OSCC and received radical surgery followed by adjuvant radiotherapy, with or without chemotherapy. Among the patients with OSCC, the subtypes were as follows: 32 patients had tongue cancer, 29 had buccal cancer, 9 had gingival cancer, 5 had retromolar cancer, 1 had floor of mouth cancer, and 14 patients had two or more OSCC subtypes. Surgical procedures for OSCC varied based on the subtype. Patients with tongue cancer or floor of mouth cancer underwent glossectomy as the primary surgery, while other OSCC subtypes underwent wide local excision for buccal, gingival, or retromolar tumors. All surgically treated patients received a reconstructive operation. All the surgeons had more than 5 years of experience in head and neck surgery for oral cancers.

The results of a univariate analysis examining the relationship between FACT-H&N scores and sociodemographic or clinical variables for all patients are summarized in Table 1. Patients with NPC had significantly better HRQoL compared to those with OSCC across several measures. Specifically, patients with NPC scored higher on the FACT-G, FACT-H&N, and three subscales of FACT-H&N (SWB, FWB, additional H&N-specific concerns), as well as TOI (Figure 1). Employment status also correlated with better HRQoL compared to employed patients across multiple measures, including the FACT-G, FACT-H&N, and four subscales of FACT-H&N (PWB, SWB, FWB, and additional H&N-specific concerns) (Figure 2).

We conducted additional univariate analyses for the two different cancer types: NPC and OSCC. The results of the univariate analysis examining the relationship between FACT-H&N scores and various sociodemographic or clinical variables in patients with NPC were summarized in Table 2. No significant differences in HRQoL were observed based on age, gender, partnership status, education level, treatment modality, and time since irradiation among the patients with NPC. However, employed patients with NPC reported significantly better PWB compared to unemployed patients. Despite this, no significant differences were detected in overall HRQoL scores as measured by FACT-G or FACT-H&N between employed and unemployed patients with NPC.

The results of univariate analyses to explore the relationship between FACT-H&N scores and various sociodemographic or clinical variables in patients with OSCC were summarized in Table 2. Older patients with OSCC (age > 50 years) exhibited significantly better EWB compared to younger patients, though no significant differences were found on FACT-G or FACT-H&N scores. Gender, education level, time after irradiation, glossectomy, and mouth angle preservation did not show significant differences in HRQoL. Surgical factors significantly impacted HRQoL: patients undergoing facial skin sacrifice had worse HRQoL in SWB and FWB, those who did not undergo inferior maxillectomy had better FWB, and those who did not undergo mandibulectomy had better HRQoL across FACT-G, FACT-H&N, FWB, and the TOI. Partnered patients with OSCC had better HRQoL, reflected in higher SWB, FACT-G, and FACT-H&N scores. Employment status was also a strong predictor, with unemployed patients with OSCC reporting worse HRQoL across FACT-G, FACT-H&N, and three subscales of FACT-H&N (SWB, FWB, and additional concerns-H&N), as well as TOI.

The results of multivariable analysis for FACT-H&N were summarized in Table 3. Across all patients, cancer type and employment status emerged as significant independent predictors of health-related quality of life (HRQoL). Specifically, patients with NPC and those who were employed reported higher HRQoL scores compared to those with OSCC and unemployed patients. Among patients with OSCC, partnership status, full-time employment, and segmental mandibulectomy were identified as independent predictors for FACT-H&N. Patients with a partner and those with full-time jobs exhibited better HRQoL. Patients who underwent segmental mandibulectomy had lower HRQoL scores (Figure 3).

## 4. Discussion

Survivorship in HNC patients presents complicated challenges, necessitating a holistic approach that focuses on both patients and their caregivers. Advancements in treatment have led to improved survival rates, making it essential to effectively address post-treatment issues to enhance our patients’ HQoL [15,16]. Understanding the factors that affect HRQoL is essential for organizing efforts towards comprehensive survivorship care. Such an approach will ensure that the physical, emotional, and social needs of HNC survivors are met, ultimately improving their overall well-being and quality of life.

Head and neck cancer significantly impacts health-related quality of life (HRQoL), with variations among different subsites such as OSCC and NPC. OSCC often leads to difficulties in eating, speaking, and facial disfigurement, which affect communication and social interactions, especially after radical treatment with combination of destructive surgery and radiotherapy [17]. Patients with NPC face unique challenges such as nasal obstruction, hearing loss, and neurological deficits due to tumor proximity to critical structures. Treatments like radiation therapy can exacerbate symptoms, and the psychological burden of diagnosis and long-term treatment further effects the impact HRQoL [18,19]. Recognizing these distinctions is vital for providing tailored care that addresses the diverse needs of HNC patients. For example, Melissa H. et al. found differences in the clinicodemographic characteristics of HPV-related vs. HPV-unrelated patients [20]. They found that different pathology types had different psychosocial outcomes after a longitudinal study. Another study did not find significant HRQoL differences among various HNC cancer survivors but noted differences in specific head and neck concerns, such as dry mouth and cough [3].

In this study, we found significant differences in FACT-G, FACT-H&N, and three subscales of FACT-H&N (SWB, FWB, additional H&N-specific concerns), as well as TOI scores between NPC and patients with OSCC, with patients with NPC having better scores. These variations may be due to differences in HRQoL measurement tools or other factors that warrant further investigation.

Understanding these factors is crucial for organizing multidisciplinary efforts towards comprehensive survivorship care, ensuring that the physical, emotional, and social needs of HNC survivors are effectively addressed to improve their overall well-being and quality of life.

### 4.1. Impact of Partnership in HRQoL

Partnerships are important for head and neck cancer patients, offering emotional support, practical assistance, and advocacy. This support is particularly crucial for patients living in rural areas or for younger patients who may face unique challenges [21,22]. Spouses or family members serve as crucial emotional anchors, aiding mental well-being and assisting with daily tasks. They also play a critical role in facilitating communication with healthcare providers, supporting lifestyle adjustments, and fostering camaraderie through shared experiences. These partnerships enrich patients’ lives, thereby enhancing their HRQoL. It is recommended that primary care clinicians should encourage the inclusion of caregivers, spouses, or partners in the usual HNC survivorship care and support, as recommended by the American Cancer Society’s head and neck cancer survivorship care guideline [23]. In this study, we found that partnerships have a significant impact on HRQoL in OSCC but not in NPC survivors. The existence of patients with OSCC with worse FWB and H&N concerns may imply that OSCC survivors need more companionship and support to improve their overall HRQoL.

### 4.2. Surgery for Oral Cavity Cancer Patients

Oral cancer surgery can significantly impact the HRQoL of individuals, affecting various aspects such as speech, eating, and overall emotional well-being. The surgical procedures involved in treating oral cancer, including tumor removal and reconstruction, may lead to changes in facial appearance, difficulties in chewing and swallowing, and potential speech alterations. These physical changes, coupled with the emotional toll of battling cancer, can profoundly impact a patient’s quality of life. In a previous study, radical surgery for HNC patients was found to have a significant impact on body image dissatisfaction. Furthermore, the surgically treated patients who had surgical procedures with facial bone destruction have even worse body image outcomes [24]. In this study, we revealed that surgery, particularly mandibulectomy, had a strong influence on the patients’ HRQoL. Mandibulectomy, the removal of part/all of the lower jaw, can lead to complications like osteoradionecrosis, malocclusion, facial disfigurement, speech and swallowing difficulties, chronic pain, and psychosocial impact [25]. These findings underscore the necessity for comprehensive support from healthcare providers to manage these issues and improve function and quality of life. Therapies and adjustments in diet and dental care are critical interventions for these patients.

### 4.3. Employment Impact to HRQoL

Kim et al. demonstrated that cancer patients generally exhibit lower HRQoL compared to the general population. Specifically, unemployed cancer patients tend to have even lower HRQoL scores than their employed counterparts [26]. Unemployment increases vulnerability to reduced HRQoL, compounded by economic survivorship stressors like treatment type, employment status, and partnership dynamics. These stressors worsen the physical and psychological effects of cancer treatment, diminishing survivors’ HRQoL. Financial stress, linked to poorer health and well-being, particularly impacts low-income cancer survivors [27]. Cancer-related financial strain affects psychological well-being and family dynamics, emphasizing the importance of addressing economic challenges in cancer survivorship [28,29]. Our study also found that employment is the most critical factor for NPC and patients with OSCC. Previous studies also demonstrate that socioeconomic status is an important factor in determining the HRQoL for HNC survivors [3,30]. Head and neck cancer survivors facing limitations localized to the head and neck region, without affecting the rest of their body or extremities, often demonstrate flexibility and determination in maintaining their professional careers and income levels. Through specialized training, these survivors can effectively manage their unique challenges, allowing them to continue contributing to the workforce and sustaining their livelihoods. By focusing on enhancing skills relevant to their job roles and leveraging support systems, these individuals showcase the power of adaptation and perseverance in overcoming obstacles post-surgery. 

The cross-sectional design of this study hinders our ability to establish causal relationships between variables, highlighting the need for longitudinal research to discern the temporal nature of these associations. With a sample size of only 150 HNC patients, limited to those with NPC and OSCC, and a 9.6% refusal rate, this study may not fully capture the diversity within this population. This limitation potentially restricts the generalizability of the findings to a broader demographic. The reliance of patients from a single medical center further raises concerns about the generalizability of the results to other settings or populations. Moreover, utilizing self-reported data introduces the possibility of recall bias, which may affect the accuracy and reliability of the information collected. While this study predominantly employs statistical analyses to explore HRQoL outcomes, it may overlook individual nuances and unique experiences that could significantly influence these outcomes. 

## 5. Conclusions

Cancer type and employment status emerged as significant independent predictors of HRQoL in patients with both NPC and OSCC. In additional to these factors, among patients with OSCC, partnership status, full-time employment, and undergoing segmental mandibulectomy were identified as independent predictors. These findings underscore the importance of developing tailored strategies to address the distinct needs of HNC survivors to improve their HRQoL. This approach will ensure that interventions are not only effective but also sensitive to the unique experiences and challenges faced by different subgroups of this patient population.

## Figures and Tables

**Figure 1 cancers-16-03366-f001:**
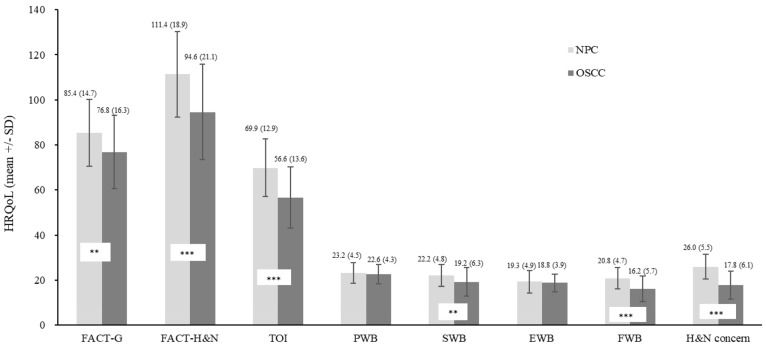
Impacts of Cancer Type on Health-Related Quality of Life (HRQoL). NPC patients had significantly better HRQoL than OSCC patients on FACT-H&N, TOI, FWB, and H&N concerns (NPC > OSCC, *p* < 0.001) and FACT-G and SWB (NPC > OSCC, *p* = 0.001). ** <0.01; *** <0.001.

**Figure 2 cancers-16-03366-f002:**
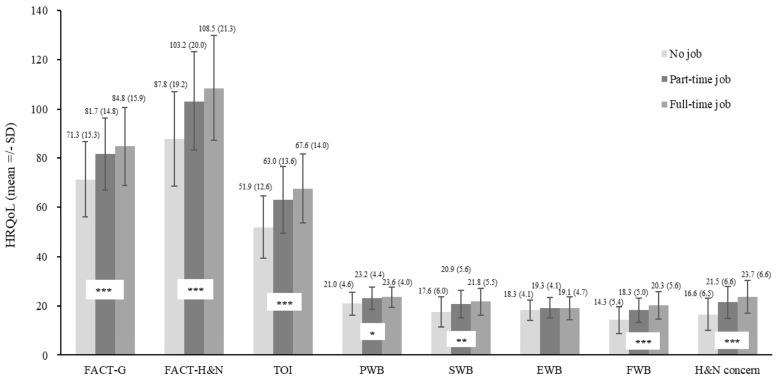
Impacts of Employment Status on Health-Related Quality of Life (HRQoL). Unemployed patients had significantly worse HRQoL on FACT-G, FACT-H&N, TOI, FWB, and H&N concerns (Part-time job, Full-time job >No job, *p* < 0.001), PWB (Full-time job >No job, *p* = 0.012), and SWB (Part-time job, Full-time job >No job, *p* = 0.002). * <0.05; ** <0.01; *** <0.001.

**Figure 3 cancers-16-03366-f003:**
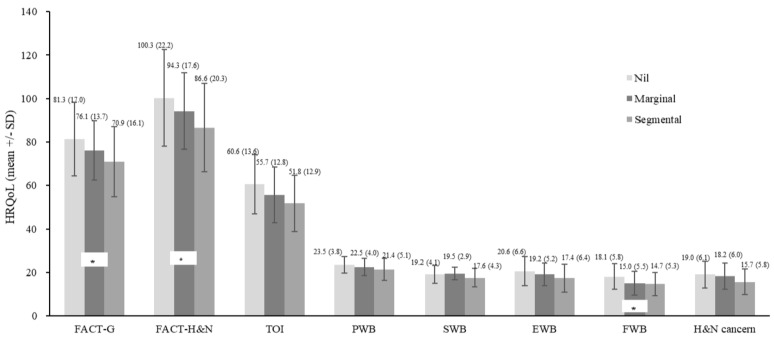
Impacts of Surgical Treatment on Health-Related Quality of Life (HRQoL). Surgical-treated patients with segmental mandibulectomy had significantly worse HRQoL than no surgical-treated patients on FACT-G (Nil > Segmental, *p* = 0.036), FACT-H&N (Nil > Segmental, *p* = 0.033), and FWB (Nil > Segmental, *p* = 0.029). * <0.05.

**Table 1 cancers-16-03366-t001:** Sociodemographic and clinical data.

Variable	N (%)	Cancer Type		
		All (n = 150)	NPC (n = 60)	OSCC (n = 90)
**Sociodemographic**				
Age (years)	≤50	69 (46.0)	31 (51.7)	38 (42.2)
	>50	81 (54.0)	29 (48.3)	52 (57.8)
Gender	Male	128 (85.3)	45 (75.0)	83 (92.2)
	Female	22 (14.7)	15 (25.0)	7 (7.8)
Partnership	No	32 (21.3)	15 (25.0)	17 (18.9)
	Yes	118 (78.7)	45 (75.0)	73 (81.1)
Employment status	No job	38 (25.3)	5 (13.8)	33 (36.7)
	Part-time job	55 (36.7)	24 (40.0)	31 (34.4)
	Full-time job	57 (38.0)	31 (51.7)	26 (28.9)
Education	Primary	27 (18.0)	8 (13.3)	19 (21.1)
	Junior high	43 (28.7)	17 (28.3)	26 (28.9)
	Senior high	48 (32.0)	16 (26.7)	32 (35.6)
	University & above	32 (21.3)	19 (31.7)	13 (14.4)
**Clinical condition**				
Time after treatment	Within two years	66 (44.0)	21 (35.0)	45 (50.0)
	More than two years	84 (56.0)	39 (65.0)	45 (50.0)
Treatment #	RT	44 (29.3)	10 (16.7)	34 (37.8)
	CCRT	106 (70.7)	50 (83.3)	56 (62.2)
Surgery	No	60 (40.0)	60 (100)	0 (0.0)
	Yes	90 (60.0)	0 (0.0)	90 (100)
**Surgical approach for OSCC**				
Glossectomy	Other surgery *			46 (51.1)
	Partial			25 (27.8)
	Total			19 (21.1)
Facial skin	Preserved			48 (53.3)
	Sacrificed			42 (46.7)
Mouth angle	Preserved			64 (71.1)
	Sacrificed			26 (28.9)
Mandibulectomy	Nil			39 (43.3)
	Marginal			24 (26.7)
	Segmental			27 (30.0)
Maxillectomy	Nil			66 (73.3)
	Inferior			24 (26.7)

NPC = nasopharyngeal carcinoma; OSCC = oral cavity squamous cell cancer; # one patient will receive more than one treatment modality; RT = radiotherapy; CCRT = concurrent chemoradiation; * Other surgery: the patients who received wide local excision for buccal/gingival/retromolar tumor but no glossectomy.

**Table 2 cancers-16-03366-t002:** Univariate analysis between FACT-H&N and sociodemographic or clinical variables.

Variable	t/f/z/χ^2^	FACT-G	FACT-H&N	PWB	EWB	SWB	FWB	H&N	TOI
**Sociodemographic**									
Age (years)	All (t)	0.11	0.46	−0.50	−1.53	0.44	1.37	1.19	0.96
	NPC (t)	0.57	0.78	0.61	0.14	−0.04	1.08	1.15	1.11
	OSCC (t)	−0.67	−0.53	−1.31	−2.48 *	0.22	0.49	−0.03	−0.22
Gender	All (t)	0.47	1.06	−0.05	−1.29	1.48	1.23	2.23 *	1.53
	NPC (z)	−0.49	−038	−0.79	−0.97	−0.50	−0.66	−0.68	0.00
	OSCC (z)	−0.80	−0.66	−0.31	−0.14	−1.32	−1.05	−0.75	−0.65
Partnership	All (t)	−1.44	−1.23	−1.24	0.83	−2.51 *	−0.78	−0.48	−0.91
	NPC (z)	−0.09	0.00	−0.79	−1.16	−1.10	−0.58	−0.10	−0.12
	OSCC (z)	−2.06 *	−1.94	−1.28	−0.78	−2.61 **	−1.38	−1.64	−1.76
Employment status	All (f)	9.22 ***	12.13 ***	4.56 *	0.61	6.71 **	14.16 ***	13.58 ***	15.54 ***
	NPC (χ^2^)	3.31	2.24	5.24	0.39	0.31	3.55	0.11	2.32
	OSCC (f)	4.24 *	5.75 **	2.12	0.11	4.21 *	5.98 **	7.66 **	7.64 **
Education	All (f)	1.99	3.15 *	1.19	1.66	0.74	2.49	5.18 **	4.05 **
	NPC (χ^2^)	1.60	3.09	4.51	3.38	0.96	1.42	5.47	4.99
	OSCC (χ^2^)	1.69	2.06	0.51	1.24	1.13	1.89	4.37	2.19
**Clinical condition**									
Cancer type	All (t)	3.29 **	4.97 ***	0.79	0.73	3.27 **	5.10 ***	8.34 ***	5.99 ***
Time after	All (t)	−1.28	−1.81	−1.05	−0.46	−1.26	−1.15	−2.67 **	−2.04 *
Irradiation	NPC (z)	−0.69	−0.79	−0.89	−0.48	−0.56	−0.36	−1.26	−1.12
	OSCC (t)	−1.02	−1.25	−0.66	−1.18	−1.01	−0.49	−1.59	−1.13
Treatment:	All (t)	−2.91 **	−2.76 **	−1.92	−2.51 *	−2.09 *	−2.44 *	−1.81	−2.48 *
RT vs CCRT	NPC (z)	−0.76	−0.63	−0.81	−0.44	−0.42	−1.13	−0.24	−0.77
	OSCC (t)	−1.95	−1.49	−1.84	−2.47 *	−1.46	−0.87	0.02	−0.94
Surgery ^#^	All (t)	3.29 **	4.97 ***	0.79	0.73	3.27 **	5.10 ***	8.34 ***	5.99 ***
**Surgical approach for OSCC**									
Glossectomy (χ^2^)		2.11	1.11	0.89	0.60	4.00	2.01	0.05	0.64
Facial skin (t)		1.88	1.69	0.47	0.79	2.26 *	2.02 *	0.80	1.36
Mouth angle (z)		−1.18	−1.10	−0.03	−0.38	−1.98 *	−1.82	−0.55	−0.79
Mandibulectomy (f)		3.4 5*	3.56 *	2.02	1.84	2.28	3.80 *	2.44	3.65 *
Maxillectomy (t)		0.97	1.19	0.34	−0.62	0.87	2.01 *	1.52	1.64

* *p* < 0.05; ** *p* < 0.01; *** *p* < 0.001; NPC = nasopharyngeal carcinoma; OSCC = oral cavity squamous cell cancer; RT = radiotherapy; CCRT = concurrent chemo-radiation. ^#^ No NPC patient underwent surgery; all OCC patients underwent surgery, and the results were the same as cancer type; z from the Mann–Whitney test; χ2 from the Kruskal–Wallis test.

**Table 3 cancers-16-03366-t003:** Multiple linear regression models for FACT-H&N.

**Multiple Linear Regression Model ^a^ for All Patients (** **n = 150)**				
	Reduced model			
			95% CI	
	Estimate	*p*	Lower	Upper
Intercept	97.09	<0.001	88.37	105.80
Cancer type:				
OSCC (ref: NPC)	−11.55	0.001	−18.39	−4.71
Employment status:				
Part-time job (ref: no job)	11.13	0.009	2.80	19.46
Full-time job (ref: no job)	14.41	0.001	5.87	22.95
**Multiple linear regression model ^b^ for patients with OSCC (** **n = 90)**				
	Reduced model			
			95% CI
	Estimate	*p*	Lower	Upper
Intercept	80.84	<0.001	70.59	91.09
Partnership:				
Partnership (ref: no partner)	13.14	0.017	2.41	23.86
Employment status:				
Full-time job (ref: no job)	12.70	0.018	2.23	23.18
Surgical treatment:				
Segmental (ref: no mandibulectomy)	−11.31	0.017	−20.55	−2.07

**^a^** Excluded variables are age, gender, partnership, and education. **^b^** Excluded variables: age, gender, education, glossectomy, facial skin sacrificed, mouth angle sacrificed, maxillectomy. Abbreviations: FACT-H&N = functional assessment of cancer therapy—head and neck; OSCC = oral cavity squamous cell cancer; NPC = nasopharyngeal carcinoma; Segmental = segmental mandibulectomy.

## Data Availability

The data presented in this study are available in this article.

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
