# Peer review of "Impacts of Employment Status, Partnership, Cancer Type, and Surgical Treatment on Health-Related Quality of Life in Irradiated Head and Neck Cancer Survivors"

_cancers, 2024, doi:10.3390/cancers16193366_

Round 1
Reviewer 1 Report
Comments and Suggestions for Authors
Dear Authors,
Review of the Article
Impacts of Employment Status, Partnership, Cancer Type, and Surgical Treatment on Health-Related Quality of Life in Irradi-ated Head and Neck Cancer Survivors
Abstract: Well written. Comprehensive
Introduction:
- "Notably, these malignancies disproportionately afflict middle-aged men, a demographic typically at the peak of their productive years, rendering the socioeconomic consequences profound for both patients and their families, consequently influencing overall HRQoL." - bibliographic reference?
Material and methods:
- This cross-sectional survey study - was approved by the Institutional Review Board (IRB No: 102-0527B) of (Taiwan northern medical center). - without brackets for the name of the medical center. Please add the city where the medical center is located.
- What method did you use to select the participants? What was the calculation formula?
- I recommend detailing the answer options and the method of calculating the results.
The results below are well presented.
The discussions are well written.
Conclusions: well written and supported by results.
I think it is a very well done article.
Author Response
We thank you and the reviewers for carefully reading our manuscript and the helpful suggestions for revision. The following is a list of the point-by-point responses to the comments and recommendations made by reviewers—Mark modification and revision in red in the Word file.
About the Introduction:
Comments 1.- "Notably, these malignancies disproportionately afflict middle-aged men, a demographic typically at the peak of their productive years, rendering the socioeconomic consequences profound for both patients and their families, consequently influencing overall HRQoL." - bibliographic reference?
Response 1: Thank you for pointing this out. We agree this comment. We listed 6 new references about these issues in page 2.
About the Material and methods:
Comments 2.- This cross-sectional survey study - was approved by the Institutional Review Board (IRB No: 102-0527B) of (Taiwan northern medical center). - without brackets for the name of the medical center. Please add the city where the medical center is located.
Response 2: Delete the bracket and add “Taoyuan City.” We revised this paragraph to “This cross-sectional survey study - was approved by the Institutional Review Board (IRB No: 102-0527B) of the medical center at Taoyuan City in northern Taiwan.
Comments 3.- What method did you use to select the participants?
Response 3: Study participants were recruited from the follow-up outpatient. When the patient met the inclusion criteria, they would be referred by a radiation oncologist. The inclusion criteria were: HNC patients who had completed the entire course of treatment, were over 20 years old, could communicate verbally, had no active disease at the time of the survey, had no major psychological disease (such as schizophrenia), and had no significant disfigurement prior to the diagnosis of HNC. After the research assistant explained the purpose of the study, participants had to agree to participate. Add and modify these descriptions in the Design and Participants of the Materials and Methods section.
Comments 4. What was the calculation formula? - I recommend detailing the answer options and the method of calculating the results.
Response: To ensure the study's sample size was sufficient, a power analysis was performed using G*Power 3.17. The analysis indicated that the study had a power of 85% with an effect size set at 0.5 and a significance level of α = 0.05. Add these descriptions to the Design and Participants section of the Materials and Methods section.
In addition, on the right side of the attached Word file, the reviewer marked the suggestions that needed modification or revision.
- Regarding the Simple Summary, please try to extend the Simple Summary to 100~150 words.
Response: We rewrite the Simple Summary into
This research aims to understand how different personal and health-related factors affect the quality of life for people who have survived head and neck cancer and have been treated with radiotherapy, sometimes along with surgery. By examining 150 patients—60 with nasopharyngeal cancer and 90 with oral cavity cancer—researchers used a specific survey to gather data on how various elements like cancer type, age, gender, relationship status, education, and employment impact their well-being. They discovered that the type of cancer, treatments received, and whether the person was employed played significant roles in influencing quality of life. For instance, patients with nasopharyngeal cancer seemed to fare better socially and functionally than those with oral cavity cancer. Additionally, being unemployed generally led to a lower quality of life, while having a partner or undergoing specific surgeries also affected outcomes. These insights could help the research community better support cancer survivors and tailor treatments to improve their overall well-being.
- Regarding the Keywords, please rewrite the highlighted sentences to reduce overlaps with the published materials.
Response: I think these key words are fully represented the article that will help future researchers to find this work.
- Regarding the References,
(a). Please list the first 10 author names before adding et al.
Response: All revisions have been completed.
(b). Please cite at least 30 references if possible.
Response: We added 6 new references to let all reference to 30.
Reviewer 2 Report
Comments and Suggestions for Authors
Thank you for the oportunity to review this study on oral cancer patients and the importance to look on psycosomatic changes after radiatio and surgery.
150 patients are treated during which period of time in this single center?
Did you analyse the factor of the leading surgeon for each patient? Please present the years of surgical experiance.
Some shortening in the introdouction and discussion part are suggested. Some philosophic aspects on further studies are not neccesary.

Author Response
Comments 1: Thank you for the oportunity to review this study on oral cancer patients and the importance to look on psycosomatic changes after radiatio and surgery.
Response 1: Thanks for the reviewer’s effort.
Comments 2: 150 patients are treated during which period of time in this single center?
Response 2. We had no detailed data for the years that the patients received radical treatment so we can not report the detail.
Comments 3: Did you analyse the factor of the leading surgeon for each patient? Please present the years of surgical experiance.
Response 3: All the surgeons had more than 5 years of experience in head and neck surgery for oral cancers. We put it in page 3 line 34-35.
Comments 4: Some shortening in the introdouction and discussion part are suggested. Some philosophic aspects on further studies are not neccesary.
Response 4: We deleted the further study suggestion. Therefore, a more comprehensive approach that includes qualitative methods is recommended to complement the quantitative findings. This would provide a deeper understanding of the factors influencing HRQoL in HNC patients and enhance the clinical applicability of the research.